# Molecular Characterization and Determination of Relative Cytokine Expression in Naturally Infected Day-Old Chicks with Chicken Astrovirus Associated to White Chick Syndrome

**DOI:** 10.3390/ani10071195

**Published:** 2020-07-14

**Authors:** Luis F. Naranjo Nuñez, Silvana H. Santander-Parra, Nicolaos C. Kyriakidis, Claudete S. Astolfi-Ferreira, Marcos R. Buim, David De la Torre, Antonio J. Piantino Ferreira

**Affiliations:** 1Department of Pathology, School of Veterinary Medicine, University of São Paulo (USP), Av. Prof. Dr. Orlando M. Paiva 87, São Paulo CEP 05508-270, SP, Brazil; fabiann7@yahoo.es (L.F.N.N.); silvanahsp@yahoo.com (S.H.S.-P.); csastolfi@gmail.com (C.S.A.-F.); daviddelatorreduque@gmail.com (D.D.l.T.); 2Facultad de Ciencias de la Salud, Carrera de Medicina Veterinaria, Universidad de Las Américas (UDLA), Av. Jose Queri, Quito 170513, Ecuador; 3Facultad de Ciencias de la Salud, Carrera de Medicina, Grupo de Investigación en Biotecnología Aplicada a Biomedicina (BIOMED), Universidad de Las Américas (UDLA), Quito 170504, Ecuador; nikolaos.kyriakidis@udla.edu.ec; 4Biological Institute, Av. Gaspar Ricardo 1700, Bastos CEP 17690-000, SP, Brazil; marcosbuim@biologico.sp.gov.br; 5Institute for Research in Biomedicine, Central University of Ecuador, Quito CP E170201, Ecuador

**Keywords:** chicken astrovirus, white chicks, cytokines, Th1, Th2

## Abstract

**Simple Summary:**

Poultry production in Brazil is a very important economic activity that provides chicken meat and eggs to global markets. Chicken astrovirus is an enteric virus related to enteric problems in chickens of several ages, but which primarily affects young chickens. Chicken astrovirus has been associated with white chick syndrome (WCS) worldwide. The objective of this investigation is to detect and molecularly characterize chicken astrovirus (CAstV) associated with WCS, determine the macroscopic and microscopic lesions and cytokine expression in the jejunum, liver, spleen and thymus of chicks naturally infected with WCS.

**Abstract:**

White chick syndrome (WCS) is an emergent disease that affects hatchability and hatched chicks, resulting in high mortality and economic losses, and is related to chicken astrovirus (CAstV). This syndrome has been reported in several countries worldwide, and groups *A iii* and *B vi* of CAstV have been determined; however, in Brazil, the virus has not been genotyped. The innate immunity of chicks affected by WCS or any CAstV is poorly understood and studied, and it is important to determine whether relative cytokine expression occurs during the early stages of the life of chicks. The aim of the present investigation is to detect and molecularly characterize CAstV associated with WCS, examine the macroscopic and microscopic lesions in the jejunum and spleen, and determine cytokine expression in the jejunum, liver, spleen and thymus of chicks naturally infected with WCS. To do so, we applied a pathological and molecular approach for CAstV detection and characterization, as well as the quantification of the relative mRNA expression of several cytokine genes. The phylogenetic analyses of the sequences obtained herein classified CAstV as uniquely belonging to group *B iv*, showing a high similarity of nucleotides (NT) (75.7–80.6%) and amino acids (AA) (84.2–89.9%) with the members of group *B* and a low similarity of NT (46.7–47.9%) and AA (37.8–38.9%) with the virus belonging in group *A*. CAstV was also detected and quantified in the serum, spleen, thymus and jejunum, the latter being the organ where CAstV had the highest viral concentration. However, this organ did not present any microscopical alterations. In contrast, we observed necrotic hepatitis in the liver of the affected subjects. On the other hand, we observed the activation of several T helper 1 (Th1)- and T helper 2 (Th2)-cytokines (*IFN-γ, IL-2, IL-8, IL-12p40, IL-15, TGF-β4, TNF-SF-15* and *t-BET*), without being able to control the viral replication due to the high concentration of viral particles in some organs, principally in the gut. One possible role of these cytokines is contributing to the control of inflammation and cell protection of intestinal cells, principally during the early activation of immune responses. However, the fact that these responses are not mature enough to control the viral infection means that more studies need to be carried out to elucidate this topic.

## 1. Introduction

Chicken astrovirus is a virus related to enteric diseases in chickens of several ages that primarily affects young animals [1,2]. This virus has been detected in Europe [3,4], Asia [5], Africa [6], India [7] and North America [8,9] and only in Brazil in South America [2,10,11].The name chicken astrovirus (CAstV) originates from the virus shape, i.e., astro means “star”. This virus is a small RNA virus of less than of 35 nm and a genome length of approximately 7.5 kb of RNA that encodes three open reading frames (ORFs), nonstructural proteins (*ORF 1a*—a protease and *ORF 1b*-RNA dependent RNA polymerase) and a structural protein (*ORF2*) that codes capsid proteins. The most variable region that is used for CAstV genotyping is a short UTR at 5’ and 3’ with a poly A tail [12,13,14,15]. CAstV is well known for its association with enteric problems [9] and problems related to kidney diseases and is characterized by gout, nephrite, and nephrosis [7], locomotion disorders [16] and hatching problems [17]. Recent studies described weak and sick animals showing the absence of feather pigmentation, resulting in chicks covered with white feathers and pale beaks, claws, and mucosa, and CAstV was identified as the unique etiological agent [8,18,19]. These animals die within a few hours after hatching or do not hatch, reducing the hatchability from 29 to 68% [17]; the animals are weak, apathetic and somnolent, with ruffled feathers a wet cloacal region; this condition in chickens is called white chick syndrome (WCS) [17] and occurs by vertical transmission, principally affecting broiler chickens [17,19]. This disease is reproduced by inoculating the virus into embryonated chicken eggs, resulting in the reproduction of the syndrome [19], which has a high mortality rate and is associated with problems in hatching; both alive and dead hatched chicks show the features of WCS, and the animals die a few hours after hatching, with a mortality of 100% [11,17,18]. Continuous outbreaks of WCS have been reported in Canada [8], Brazil [11] and in some countries in Europe [17] where CAstV has been detected. The gene *ORF2* of viral capsid is used for CAstV genotyping [3]. To date, two groups of CAstV have been identified, *A* and *B*, and the hypervariability of *ORF2* has added many subgroups within both groups [3]; the first report of WCS showed the macroscopic features of this syndrome in animals in Ireland, Finland, Norway, and the United Kingdom and classified CAstV into Group *B* subgroup *vi* [3,15]; however, CAstV causing WCS in Poland was characterized as Group *A* subgroup *iii* [20]. Regarding cases of WCS in Ontario, CAstV was also characterized into Group *B ii* [8], showing that the virus related to WCS has high genetic variability; nevertheless, information regarding the genetic features of CAstV associated with WCS in Brazil is lacking. The pathological features observed in animals with WCS mainly include alterations in the liver, which exhibits hepatomegaly with green to yellow foci, necrosis in hepatic cells and intestines filled with yellow to green liquid content with the presence of bubbles of gas, but microscopic alterations have not been reported in association with this syndrome [18]. However, experimental infections with CAstV associated with enteric problems have resulted in microscopical cystic enteritis [9]. Furthermore, the immunological aspect of CAstV infection is poorly studied, and information related to this virus is lacking. Experimental models using poults to study enteritis have shown that Turkey Astrovirus 2 (TAstV-2) produces enteritis, while simultaneously inducing the expression of the mRNA of the *TGF-β* gene [21]. Experimental infections with human astrovirus (HAstV) in Caco2 cells increased cell barrier permeability and induced *IFN-β* upregulation, which lead to viral replication inhibition by blocking the expression of positive-strand viral RNA and capsid protein synthesis [22]. Currently, information regarding the immunological aspects of infection with CAstV is lacking, and knowledge of such important aspects could enhance our understanding of the viral pathogenicity and could be used for developing a vaccine and tools for disease control. These aspects are important and should be continuously studied to understand the relationship between CAstV and WCS; thus, the aim of the present work was to genetically characterize CAstV in chicks with WCS, determine the macroscopic and microscopic lesions present in many organs of infected chicks, detect and quantify the virus in tissues and determine whether relative cytokine expression occurs in the liver, spleen, jejunum and thymus of chicks affected by WCS.

## 2. Materials and Methods

### 2.1. Chicks, Postmortem and Histopathological Examination

In the present work, we used ten one-day-old chicks with white chick syndrome (WCS), which were obtained from incubators, and presented high mortality and impairment within the first week of age. All chicks were housed at the Laboratory of Avian Diseases from School of Veterinary Medicine—University of São Paulo, Brazil. The chicks were individuality weighed and subjected to molecular and pathological analyses. The chicks were slaughtered in accordance with the guidelines and the approval of the Committee on the Care and Use of Laboratory Animal Resources of the School of Veterinary Medicine, University of São Paulo, Brazil, under number #2569/2012, and a postmortem examination was carried out to examine the macroscopic abnormalities, such as white fluff, beaks and pale legs. After the opening of the celomatic cavity, the organs were examined, and the presence or absence of macroscopic lesions was determined. From each chick, we collected the liver, jejunum, thymus, spleen, and blood to obtain serum. Each sample was stored according to the analysis that was subsequently performed. A fragment of the liver or intestine (jejunum) was collected and fixed in buffered 10% formalin, pH 7.0; subsequently, the fragments were proceeded, paraffined, sectioned into 5 µM sections, fixed and stained with hematoxylin and eosin, and analyzed under light microscopy.

### 2.2. Procedure for Molecular Analyses

To proceed with the molecular analyses, the harvested organs were weighed and processed as described below. A fragment of approximately 50 mg of the liver, 50 mg of the jejunum, and the whole spleen and thymus, were placed individually in 2-mL Eppendorf microtubes and sterilized and deposited into liquid nitrogen; other parts of the liver and intestine were macerated and stored at −80 °C. The serum was obtained, and 250 µL was placed in a 2-mL sterile microtube. The total RNA extracted from these samples was tested to determine the presence or absence of CAstV. Then, the viral load and relative expression of cytokine genes in each organ, except for the serum, were quantified through qPCR. Intestine samples positive for CAstV were used for the molecular characterization of the *ORF2* gene and its genotyping.

### 2.3. CAstV Detection and the Quantification of the Relative Expression of Cytokine Genes

For the detection and molecular characterization, an intestine fragment was placed in a sterile microtube containing phosphate buffered saline (PBS) 0.1 M, pH 7.4. The samples were frozen at −80 °C for 1 min, thawed for 1 minute at 56 °C and homogenized. This procedure was repeated three times, and then, the samples were centrifuged at 12,000× *g* for 30 min Afterwards, an absolute quantification of viral RNA and relative quantification of the expression of messenger RNA, 50 mg of the liver or jejunum or whole spleen or thymus were disrupted using Tissue Lyser LT (Qiagen, Redwood city, CA, USA.) and immediately subjected to RNA extraction.

### 2.4. RNA Extraction

RNA was extracted from 250 µL of a supernatant of viral suspension obtained as described above (molecular characterization), 250 µL of serum, 50 mg of organs (liver and jejunum) and whole organs in the case of the thymus and spleen using TRIzol reagent (Invitrogen by Life Technologies, Carlsbad, CA, USA) according to the manufacturer’s instructions. RNA was treated with DNase using a DNA-free DNA Removal Kit (Invitrogen) according to the manufacturer’s instructions.

### 2.5. Real-Time RT-PCR Assay for the Detection and Quantification of CAstV

In the present study, RT-qPCR was performed for the detection and absolute quantification of CAstV based on the ORF 1b gene. For this purpose, a reaction of endpoint PCR previously described by Day et al. [23] was used. The extracted RNA (1 µg) was subjected to reverse transcription using a Super Script First-Strand Synthesis System according to the manufacturer’s instructions, using 1 µL of oligo (dT)_20_ and 1 µL of random hexamer primer. The obtained cDNA was subjected to an endpoint PCR. After the PCR amplification, the generated product was inserted into a PCR 2.1-TOPO vector (Invitrogen) and transformed and cloned into *E. coli* competent cells according to the manufacturer’s instructions. Plasmid DNA was extracted from a culture of clone bacteria using a QIAprep Spin Miniprep Kit (Qiagen). The plasmid DNA was subjected to endpoint PCR to confirm that the PCR fragment was inserted into the vector and sequenced three times in both directions using a Big Dye Terminator Version 3.1 Cycle Sequencing Kit (Invitrogen). The sequencing reaction was carried out using an ABI 3730 DNA Analyzer (Invitrogen). The obtained sequences were edited and aligned using CLC Main Workbench 7.0.2, and the similarity with other CAstV sequences present in GenBank was determined using the BLAST tool. Based on the obtained sequences, a pair of primers was designed using the software package Geneious 11.1.5 (Table 1).

### 2.6. RT-qPCR Based on SYBR Green

The cDNA obtained from the samples, as described above, was subjected to real time PCR. The reactions were performed using a mixture containing 10 µL of Fast SYBR Green Master Mix (2X) (Invitrogen), 0.5 µM of each primer (Table 1), 1 µL of cDNA from each sample and UltraPure™ DNase/RNase-Free Distilled Water dH_2_O (Invitrogen) necessary to complete 20 µL. No template control (NTC) was used, substituting the cDNA with water dH_2_O. The reaction was performed using a 7500 fast real time PCR system (Invitrogen) in the fast mode using the following steps: 95°C for 5 min, 40 cycles of 95 °C for 15 s and 60 °C for 30 s. The melting curve was generated by heating at 95 °C for 10 s, followed by lowering the temperature to 60 °C for 1 min and heating to 95 °C. To determine the sensibility of the assay, plasmid DNA extracted from the bacterial culture that contained the PCR fragment of the ORF 1b gene of CAstV was quantified, and the curve was generated based on ten serial dilutions with a base of 10, resulting in a standard curve with 10 to 10^9^ plasmid copy numbers. The limit of detection (LOD) and limit of quantification (LOQ) were determined. The efficiency of qPCR was also determined with the plasmid serial dilution method. The absolute quantification of CAstV was performed based on the standard curve, which was generated here and presented as viral gene copies per mg of tissue. 

### 2.7. Molecular Characterization of the ORF2 Capsid Gene of CAstV

The molecular characterization of CAstV was carried out with the jejunum samples. For this purpose, an endpoint RT-PCR reaction was used as described [3]. Then, 1 µg of total RNA was subjected to an RT reaction as described above. The obtained cDNA was submitted to a PCR containing the following: 0.5 µM of each primer (Table 1), 2.5 µL of buffer 10×, 4 µL of dNTPs 1.25 mM, 37.5 mM of MgCl_2,_ 1.0 U of Platinum Taq DNA polymerase and 2 µL of cDNA. The reaction was performed at the same temperatures previously described [3]. For the sequencing of the amplified cDNA of the *ORF2* gene, the amplified product was purified using CleanSweep PCR Purification (Thermo Fisher, MA, USA) as described by the manufacturer, inserted into a PCR^TM^ 2.1-TOPO vector (Invitrogen), and transformed and cloned into *E. coli* competent cells according to the manufacturer’s instructions. The plasmid DNA was subjected to endpoint PCR to confirm that the PCR fragment was inserted into the vector and sequenced three times in both directions using a Big Dye Terminator Version 3.1 Cycle Sequencing Kit (Invitrogen). To flank the full-length sequence, the primer “walking strategy” was used along with the M13 Forward and Reverse primers to obtain a 2.2 Kb sequence. The sequencing reaction was carried out using an ABI 3730 DNA Analyzer (Invitrogen). The obtained sequences were edited and assembly using the de novo assembly method, the ORF was predicted using de Geneious software package 11.0.1 (Biomatters Ltd., Auckland, New Zealand), and complete Coding Sequence (CDs) were obtained. The generated sequences were analyzed using the BLAST tool to determine the similarity between the sequences and other sequences deposited in GenBank. The nucleotide sequences obtained here were aligned and compared with other sequences of CAstV from other parts of the world using the CLUSTAL W method available in the ClustalX 2.0.11 software package (European Bioinformatics Institute Saffron Walden CB 10 1SD, UK). The phylogenetic analyses were performed using a Neighbor-joining statistics method along with a p-distance substitution model and phylogeny test bootstrap model with 1000 replicates integrated in the MEGA version 7 software package [33].

### 2.8. RT-qPCR for the Quantification of the Relative Expression of Cytokine Genes

For the detection, determination and relative quantification of cytokine genes belonging to T helper 1 (Th1) and T helper 2 (Th2) [24], the immune response to *IFN-γ* [24], *IL-2* [26], *IL-8* [27], *IL-12* p40 [28], *IL-15* [29], *TGF-β4* [31], *TNF-SF15* [30] and *t-BET* [25] and, as the endogenous control, *β-Actin* [32], RT-qPCR reaction was used for each gene. The primers used in the RT-qPCR assays for the amplification of the cytokine genes are listed in Table 1.

The cDNA was obtained from the samples as previously described. The cDNA was diluted to a 1:10 concentration and used in the qPCR reactions. The RT-qPCR reactions were performed with a final volume of 20 µL. Each reaction had 10 µL of PowerUp^TM^ SYBR^®^ Green Master Mix (2X) (Thermo Fisher), 0.5 µM of the forward and reverse primers each, 11 µL of cDNA from each sample and UltraPure™ DNase/RNase-Free Distilled Water dH2O (Invitrogen) until complete 20 µL. The reactions were performed using a 7500 Fast Real-Time PCR System (Thermo Fisher) using the following conditions: preincubation to 95 °C for 10 min, followed by 45 cycles of 95 °C for 15 s, 58 °C –64 °C for 15 s and 72 °C for 1 min The analysis of the melting curve was performed in three steps as follows: 95 °C for 10 s, followed by a decrease in temperature to 58 °C for 1 min, and heating to 97 °C. A housekeeping gene (*ß-Actin*) was used for the relative gene expression analyses. Relative gene expression analyses were carried out using the 2DDCt method described previously [34]. The data represent the mean of 5 biological replicates (chickens).

### 2.9. Statistical Analysis

The difference in the weight of each animal and each organ analyzed in the present study was determined using the Wilcoxon rank sum test, and the difference in the absolute viral quantification in each tissue analyzed was inferred using the Kruskal–Wallis rank sum test; both tests were calculated using the R software package.

## 3. Results 

### 3.1. Postmortem Examination of the Chicks and Histopathology 

In the present work, ten chicks with WCS were analyzed; their whole body was covered with white fluff and the beak, legs and claws were pale (Figure 1). During the necropsy of the celomatic cavity, we observed the marked presence of an enlarged liver with a brown, yellow to orange color and focal and multifocal pale yellow and green areas (Figure 1). The whole intestine was dilated and filled with yellow to aqueous green liquid content and foamy content. The unabsorbed yolk sac was presented. The spleen, thymus and remaining organs did not show any apparent macroscopic lesions. The liver of the chicks showed necrotic areas with the presence of cells undergoing degeneration and apoptosis, vacuolization, and the loss of liver parenchyma; the liver architecture was altered, showing the loss of hepatic trabeculae. The presence of heterophile surrounding the centrilobular vein and intense hepatitis in the liver zone three were also observed (Figure 2). The microscopic analysis of the jejunum did not show any alterations. 

### 3.2. Molecular Analysis

#### 3.2.1. RT-qPCR Based on SYBR Green to Test Whether CAstV Is Associated with White Chick Syndrome (WCS)

The assay performed in the present study could detect and quantify 10^9^ plasmid copies to ten plasmid copies. The LOD and LOQ were determined in ten plasmid copies. The melting curve was clean with a unique peak without any alterations with a melting temperature of 74.47 °C (Figure 3), and no dimers were observed. The standard curve was determined with an efficiency of 106.42%, a slope of −3.177 and a correlation of 0.995 (Figure 3).

#### 3.2.2. Detection and Quantification of CAstV Associated with WCS

The white chicks were positive for CAstV in the serum, spleen, liver, jejunum and thymus (Table 2), and the highest concentration of this virus was present in the jejunum (*p* < 0.05), followed by the spleen, liver, thymus and serum.

#### 3.2.3. Molecular Analysis of the ORF2 Capsid Gene of CAstV

CAstV-positive jejunum samples from five WCS samples were randomly selected, and the complete ORF 2 gene of CAstV was amplified, cloned, and sequenced, resulting in five sequences of the complete gene (2214 bp). These CDs were analyzed and compared with other sequences deposited in GenBank. These sequences were grouped together with the sequences of group *B* of chicken astrovirus; within this group, the Brazilian white chick sequences were clustered in a separate group, the *iv* group, which had a high similarity of nucleotides (NT) (99.5–99.8%) and amino acids (AA) (99.4–100%). These sequences showed a low similarity of NT (46.8–47%) and AA (38–38.9%) compared with the sequences belonging to group *A* of CAstV and a high similarity of NT (75.7–80.7%) and AA (84.6–89.9%) with sequences of CAstV belonging to group *B* of CAstV (Table 3). The phylogenetic analyses showed two well defined groups *A* and *B*; in these groups, the subgroups *i, ii, iii* were identified in group *B*. The obtained sequences were grouped into a separate group, forming subgroup v*i* with a bootstrap of 100%. Similarly, in group *A*, the subgroups *i, ii* and *iii* were identified, and the last subgroup included the sequences of white chicks from Europe (Figure 4). The accession number of ORF2 sequences of CAstV obtained here are as follows: USP 748-12 (MN329809.1), USP 748-14 (MN329810.1), USP 748-16 (MN329811.1), USP 748-18 (MN329812.1), and USP 748-20 (MN329813.1). Figure 4 shows the sequences of the CAstV associated with WCS in Brazil.

### 3.3. Quantification of the Relative Expression of Cytokine Genes

The relative expression of cytokine genes was measured in the jejunum, liver, spleen and thymus of birds naturally infected with CAstV and showing WCS and was normalized against the relative expression of the same genes of the non-affected group. The cytokine genes of interest were *IFN-γ*, *IL-2*, *IL-8*, *IL-12 p40*, *IL-15*, *TGF-β4*, *TNF-SF-15* and *t-BET* (Figure 5). The relative expression of mRNA of genes *IL-2*, *IL-8*, *IL-15* and *TNF-SF-15* was determined in all four tested organs, whereas the relative expression of *IL-12 p40* was measured in all organs but the thymus, where no expression was found. *TGF-β4* and *t-BET* were only expressed in the spleen, jejunum, and liver. Finally, measurable *IFN-γ* gene expression was only observed in the thymus and liver. Overall, of the four organs analyzed, only in the liver was it possible to detect and measure the relative expression of all cytokine genes (Figure 5). These results indicate that there is a relative expression of the mRNA genes of cytokines at different levels in the four analyzed organs in animals affected with WCS at one-day old, showing that the virus could be the responsible for these immunological aspects when it replicates in the organs.

## 4. Discussion

Chicken astrovirus (CAstV) is important for poultry health because it causes high economic losses related to the presentation of enteric, kidney and hatching problems [7,9,17] The latter problem causes high mortality in incubators with the presentation of weak chicks, showing the features of white chick syndrome (WCS), which is a syndrome caused by CAstV, leading the birds to die within the first twelve hours [8,11,19]. The WCS reduces rates of hatchability between 29 and 68% and the hatched chicks die in the first 24 hours, which results in 100% mortality, if any affected chicks stay alive [17,19]. WCS is related to CAstV group *A iii* and *B iv* [3,8,17,19], the virus has a vertical transmission, from the layers of flocks affected with either of the genotypes of the virus [17,19]. WCS has been reported in several countries in Europe [17,19], Canada in North America [8,35] and, uniquely, in Brazil and South America [11]. In Brazil, outbreaks of WCS have been continually reported in Brazilian farms, and the continuous presentation of these pathological problems makes it an important subject to study; thus, here, we performed the detection and genetic characterization of the *ORF2* gene of CAstV associated with WCS, examined the macroscopic and microscopic alterations present in chicks affected by WCS and determined the relative cytokine expression present in the jejunum, thymus, spleen and liver of these naturally affected animals.

The first report of WCS in Brazil was published by our working group in 2016 [18] and we described the pathological findings observed in postmortem examinations of thirty sick animals, including chicks covered with white fluff, with pale beaks, legs and claws, and some alterations in the liver and intestine, which were also observed in the present study and are consistent with other reports in other countries [8,17]. The histopathological findings present in the liver, such as the presence of hepatitis with several necrotic zones, are consistent with other previous reports [8,17], showing that the Brazilian CAstV causes hepatitis and could be the principal cause of death in chicks affected by WCS. 

The intestine of the analyzed chicks with WCS presented no microscopic alterations. However, the intestines of the affected animals were dilated and filled with yellow-green liquid content with gas bubbles, and were found to have the highest viral concentration (*p* < 0.05) than the rest of the organs examined, showing that the intestine could be the organ where CAstV replicates more. It is known that CAstV produces osmotic enteritis [36] accompanied by the absence of any pathological alterations in the intestinal epithelium, but a diminished nutrient absorption [21]. This could be causing the absence of yolk consumption and absorption by the gut epithelium, a condition that could already appear few days after the beginning of the egg´s incubation. 

The yolk sac provides nutrients to the embryo during its development. The yolk sac also provides carotenoids for ornament and antioxidant procedures for body maintenance, which occurs at the moment of hatching in normal yellow chicks, principally by the uptake of lutein and zeaxanthin from the yolk [37,38]; this sac decreases along with the chick until it is completely consumed [39,40]; however, as reported here, chickens affected by WCS present larger and heavier yolk sacs, showing decreased yolk consumption, resulting in weak and uncolored animals. 

Herein, we also sought to develop a RT-qPCR based on SYBR Green to amplify a part of the *ORF1b* gene. This assay showed high specificity and sensitivity for the detection and quantification of CAstV and could detect 10 viral copies in a short time; this assay could detect and quantify the virus in the serum, liver, jejunum, spleen and thymus, with high specificity and sensitivity similar to previous molecular technique used for CAstV detection and quantification [41]. As previously reported, CAstV related to WCS was detected but not quantified in several organs, more CAstV was detected in the gizzard than the other organs, followed by the intestine and lung [11], and other studies have shown the detection of CAstV in jejunum, kidney and liver [8,19]. The present study showed the quantification of CAstV in the serum, jejunum, liver, thymus and spleen, and the highest viral concentration was found in jejunum; the presence of viral particles in the serum shows the presence of viremia, probably since the early stages of chicken life during incubation, confirming the vertical transmission of CAstV [19] and explains the mortality and hatching problems [17,19]. As previously reported, viral particles are present in tissues other than the enteric system, such as the lung (93.33%), brain (10%), and heart (6.67%); however, the detection of the virus in the liver (13.13%) was minimal [18], while, in the present study, the presence of CAstV in the liver was high (100%), which could be explained by the use of the more sensitive technique of RT-qPCR.

Here, the molecular genetic features of the *ORF2* gene of CAstV associated with WCS was investigated, complete gene sequences were obtained and the molecular comparison and analysis determined that the Brazilian CAstV sequences related to WCS are grouped and belong to the *B iv* group of CAstV, which is the phylogenetic group that contains the previously classified sequences of CAstV associated with the first report of WCS [15]; in the present work, it was not possible to analyze the sequences obtained here with the last mentioned sequences due to the lack of availability in GenBank. Interestingly, in Europe, two circulating genetic groups are associated with WCS, i.e., the mentioned group *B iv* and the group *A iii* from Poland [15,19], showing that both genotypes are associated with WCS despite the low similarity of AA and NT [3]. To the best of our knowledge, this is the first report of a molecular characterization of CAstV associated to WCS in Brazil. We found that that the virus related to WCS in the present work is different to other CAstV reported in Brazil related to enteric problems and classified in the B ii group [42], shedding light on the understanding of WCS pathology and implying that different genotypes are circulating in Brazilian commercial flocks. However, more studies need to be carried out to determine the genetic spectrum of CAstV variants in Brazil.

The immunity related to CAstV is not well understood, until now, there has been limited information about how the chickens react against the virus; this lack of information makes difficult understand which cytokines are related to its infection. Relative gene expression levels of several relevant cytokines were investigated in the liver, jejunum, spleen and thymus of chickens naturally affected with WCS. The results of cytokine gene expression levels in studied organs showed the activation of several T helper 1 (Th1) and T helper 2 (Th2) response related cytokines. *IFN-γ*, *IL-2*, *IL-8*, *IL-12 p40*, *IL-15*, *TGF-β4*, *TNF-SF-15* and *t-BET* were all found to be expressed in the liver of chicks with WCS, whereas parts of them were expressed in the jejunum, spleen and thymus. Of great interest was the absence of *IFN-γ* expression in the jejunum and spleen, despite the high virus concentration in both organs. At the crossroads between innate and adaptive immunity, high *IFN-γ* expression implies already activated innate immune responses and the activation of *IFN-γ* secreting NK and Th1 cells [43,44]. These are non-acute, late immune responses. However, the affected studied animals had merely a few hours of life, so it can be speculated that this could be due to the fact that the immune responses are still immature or did not have enough time to develop [45].

Despite the yellow to green diarrhea in the analyzed chicks, we observed an absence of histopathologic changes in the intestine, accompanied by an absence of any type of immune cell infiltration. These characteristics are like the ones observed in a previous investigation where poultry infected with Turkey Astrovirus (TAstV) presented intense diarrhea with mild or insignificant microscopic lesions and the absence of inflammatory exudates [21]. In the same study, it was showed that this lack of lesions was due to the increased activation of the potent immunosuppressive cytokine transforming growth factor beta (*TGF-β*) during astrovirus infection. A significant gene expression of *TGF-β* was also found in jejunum samples from our study, potentially explaining the downregulation of inflammation and the absence of histopathologic lesions.

Additionally, the highest expressions of both *IL-8* and *IL-15* was found in the liver. In line with this, we observed significant immune cell infiltration in this organ, mainly consisting of heterophiles and, secondly, lymphocytes. *IL-15* has an anti-apoptotic effect on cytotoxic cells, therefore keeping them in play and being capable of killing a high number of infected cells in the case of viral infections. On the other hand, *IL-8* is a chemokine known to recruit heterophiles and macrophages in the site of infection, possibly contributing to the infiltrate profile we observed in the infected livers. To further support this, experimental infection with Fowl Adenovirus Type 8 produces the upregulation of *IFN-γ* and *IL-8* in the spleen and liver, with an important role in driving the immune responses against Fowl Adenovirus Group 8 (FAdV-8) infection [46].

In the liver, we also found high concentrations of *TNF-SF-15* and *IL-12 p40*. As was earlier discussed, this is an organ in which parenchymal damage was observed, along with the increased presence of necrosis, apoptotic phenomena, and high numbers of infiltrates. *TNF-SF-15* is a cytokine produced by endothelial cells acting in an autocrine fashion, resulting in decreased proliferation and inducing their apoptosis [47], thereby possibly explaining the apoptotic phenomena observed microscopically. The local expression of *IL-12 p40* is suggestive of the activation of the innate immune system against the virus, attempting to mount a Th1-mediated response. 

Infection with Fowl Adenovirus Group 4 (FAdV-4) is reported to increase the expression of *IL-6*, *IL-8* and *TNF-α* and produce tissue damage in lymphoid organs, principally through apoptosis [48]. The spleen and thymus of CAstV infected chicks need to be studied in more depth to determine whether the virus and the ensuing cytokine secretion can cause damage in these organs. Furthermore, we showed that CAstV replicates in the intestine epithelium, lymphopoietic organs and liver; however, the mechanisms underlying the generation of immune responses against CAstV are yet to be fully characterized and more studies need to be carried out in order to elucidate this topic.

To our knowledge, this the first report of early life cytokine gene expression in organs of CAstV chicks affected by WCS, demonstrating the early activation of the innate immune system of embryos and newly hatched chicks against a viral pathogen, as was previously described [45]. The present study showed that CAstV replicates in the epithelium of the intestine, lymphoid organs, and liver, but the mechanism of the generation of an innate immune response against CAstV infection in these tissues has not been fully characterized, and more studies are needed to elucidate and provide more information regarding this important topic.

## 5. Conclusions

The present study showed that CAstV replicates in the epithelium of the intestine, lymphoid organs, and liver, but the mechanism of the generation of an innate immune response against CAstV infection in these tissues has not been fully characterized, and more studies are needed to elucidate and provide more information regarding this important topic. Overall, we herein describe, for the first time, the cytokine expression profile of chicks affected with WCS, showing the activation of mainly Th1-oriented immune responses, as assessed by the expression of *IFN-γ*, *IL-2*, *IL-8*, *IL-12*, *IL-15*, *TGF-β4*, *TNF-SF-15* and *t-BET*. Following infection with CAstV, the animals were shown to display an early cytokine-mediated immune response, indicating that the development of an anti-CAstV vaccine using the attenuated virus could be possible. Such a vaccine could be administered in the infected breeder hens to prevent the vertical transmission of the virus. On the other hand, the genetic features of *ORF2* CAstV gene showed that this virus belongs to a different group than the ones responsible for kidney and enteric manifestations in chickens. Finally, we also genotyped, for the first time, the CAstV variant related to Brazilian cases of WCS. These findings drive us to continue studying CAstV associated with WCS to elucidate the problem and develop possible alternatives for its control.

## Figures and Tables

**Figure 1 animals-10-01195-f001:**
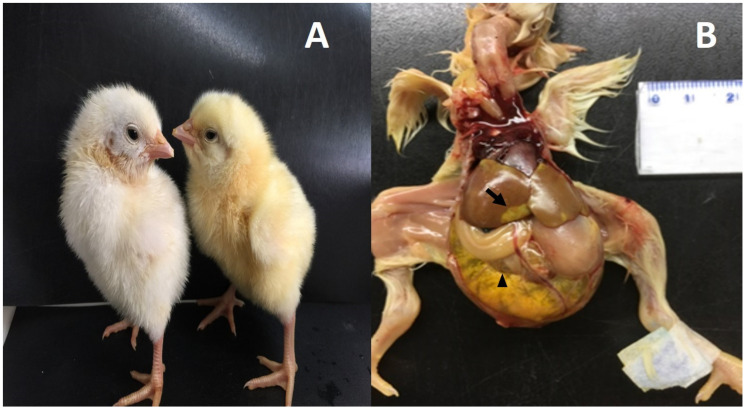
Macroscopic lesions observed in animals affected with white chick syndrome (WCS); (**A**): left: chick affected with WCS, right: non-affected chick; (**B**): macroscopic lesions present in jejunum and liver in the animals with WCS; hepatomegaly, with focal pale yellow areas (arrow), and duodenal loop filled with liquid (arrowhead).

**Figure 2 animals-10-01195-f002:**
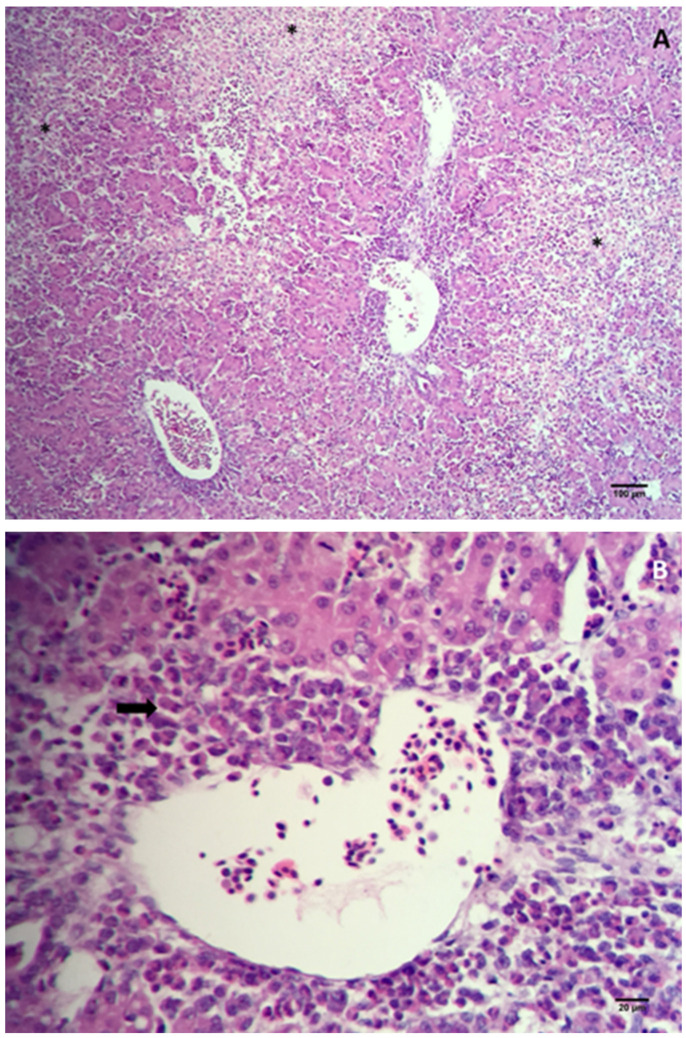
Photomicrography of liver sections from chicks with a one-day-old WCS. Liver parenchyma showed necroses—asterisks (**A**) and increases in the infiltrate of granulocytes—arrow (**B**) indicating necrosis hepatitis. All slides were stained with hematoxylin–eosin (HE).

**Figure 3 animals-10-01195-f003:**
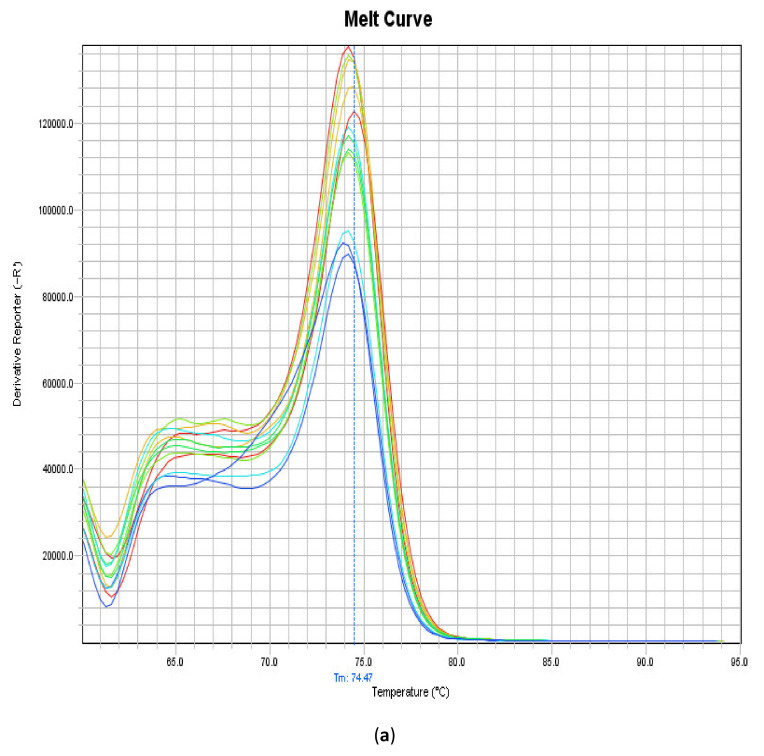
Melting (**a**) and efficiency (**b**) curves of the RT-qPCR assay performed to amplify and quantify CAstV associated with WCS. The melting curve showed a melting temperature of 74.47 °C, and the standard curve had an efficiency of 106.4%.

**Figure 4 animals-10-01195-f004:**
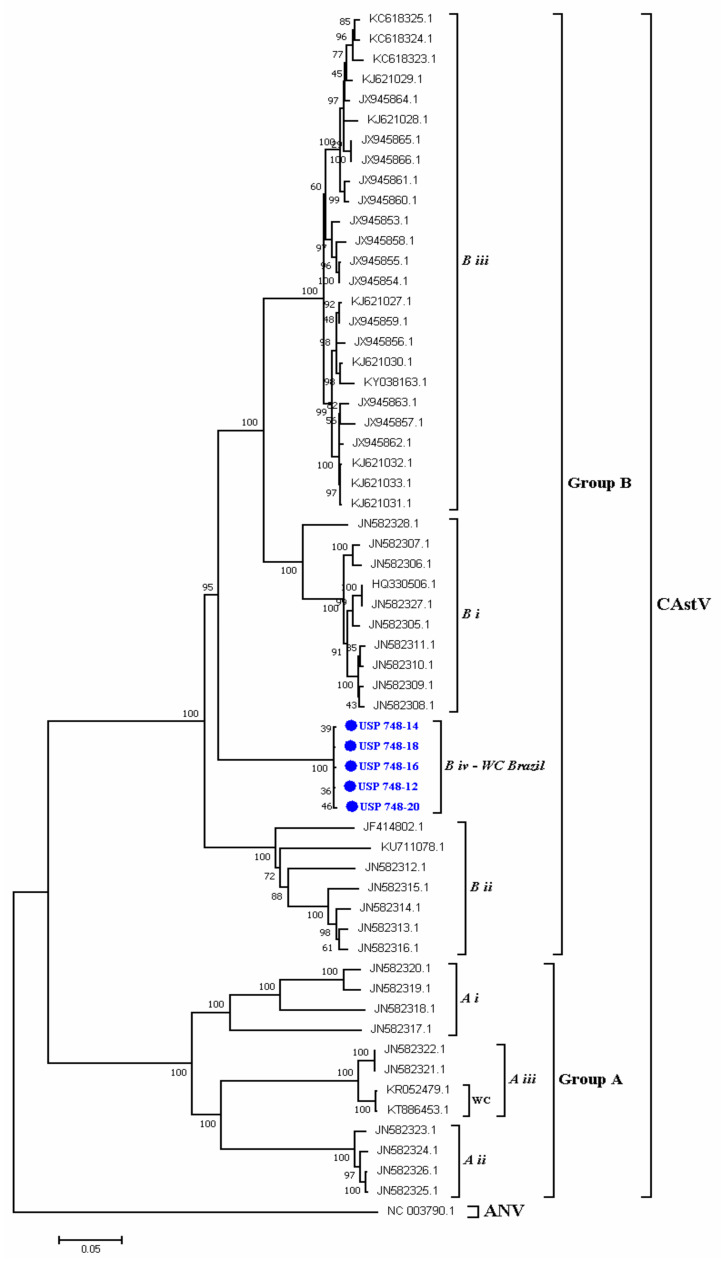
Phylogenetic relations between the sequences of CAstV belonging to chicks naturally infected with WCS obtained in the present work and other sequences of CAstV based on complete ORF2 gene nucleotide sequences. Sequences were aligned using the CLUSTAL W method in ClustaX2 2.1. The phylogenetic tree was constructed using the MEGA 7 Software Package. The numbers along the branches refer to the bootstrap values of 1000 replicates. The scale bar represents the number of substitutions per site. Avian Nephritis virus (ANV) was used as the outgroup. The sequences obtained here are shown in blue.

**Figure 5 animals-10-01195-f005:**
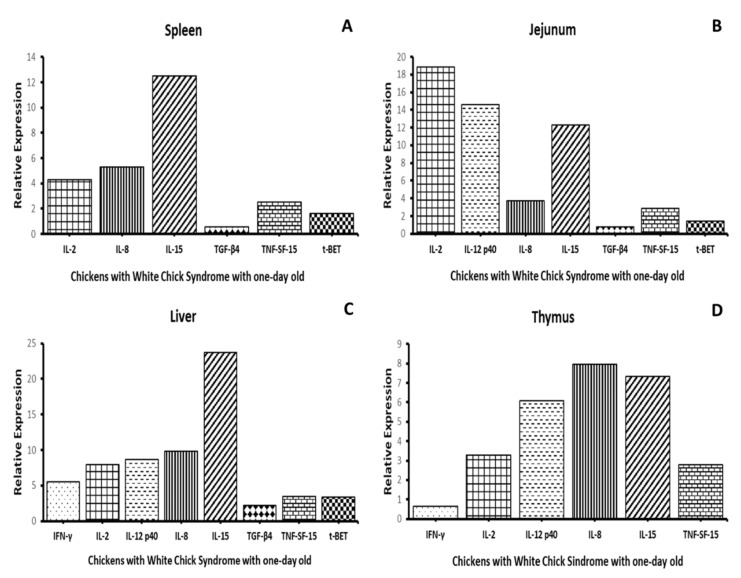
Relative gene expression of cytokines. Relative gene expression of *IFN-γ*, *IL-2*, *IL-8*, *IL-12 p40*, *IL-15*, *TGF-β4*, *TNF-SF-15* and *t-BET* in the spleen (**A**), jejunum (**B**), liver (**C**) and thymus (**D**) of chicks naturally affected with white chick syndrome. The data represent the mean gene expression levels of the target genes in five biological replicates relative to the housekeeping gene *β-actin*.

**Table 1 animals-10-01195-t001:** Sequences of primers used in the molecular assays for the detection and quantification of chicken astrovirus (CAstV) and the relative gene expression of cytokines in the spleen, jejunum, liver, and thymus of chicks naturally affected with white chick syndrome.

Virus	Primer	Target Gene	PCR Assay	Nucleotide Sequences (5′–3′)	Reference
CAstV	CAsPol1F	*ORF 1b*	RT-PCR	GAYCARCGAATGCGRAGRTTG	[23]
	CAsPol1R			TCAGTGGAAGTGGGKARTCTAC	
	CastV PRECAP	*ORF 2*	RT-PCR	TAGAGGGATGGACCGAAATATAGCAGC	[17]
	CastV POSTCAP			TGCAGCTGTACCCTCGATCCTA	
	CASTV-W-C-F	*ORF 1b*	RT-qPCR	TTGATGGCACTATTCCAAAGG	This Study
	CASTV-W-C-R			GAATCTGATCTGCCGAATGC	
Cytokine Expression		*IFN-γ*	RT-qPCR	ACACTGACAAGTCAAAGCCGC	[24]
				AGTCGTTCATCGGGAGCTTG	
		*t-BET*	RT-qPCR	GGG AAC CGC CTC TAC CTG	[25]
				AGTGATGTCGGCGTTCTGG	
		*IL-2*	RT-qPCR	TGCAGTGTTACCTGGGAGAAGTGGT	[26]
				ACTTCCGGTGTGATTTAGACCCGT	
		*IL-8*	RT-qPCR	CCAAGCACACCTCTCTTCCA	[27]
				GCAAGGTAGGACGCTGGTAA	
		*IL-12p40*	RT-qPCR	CCAAGACCTGGAGCACACCGAAG	[28]
				CGATCCCTGGCCTGCACAGAGA	
		*IL-15*	RT-qPCR	TCTGTTCTTCTGTTCTGAGTGATG	[29]
				AGTGATTTGCTTCTGTCTTTGGTA	
		*TNFSF15*	RT-qPCR	CCTGAGTATTCCAGCAACGCA	[30]
				ATCCACCAGCTTGATGTCACTAAC	
		*TGF-β4*	RT-qPCR	CGGCCGACGATGAGTGGCTC	[31]
				CGGGGCCCATCTCACAGGGA	
		*β-Actina*	RT-qPCR	CAACACAGTGCTGTCTGGTGGTA	[32]
				ATCGTACTCCTGCTTGCTGATCC	

**Table 2 animals-10-01195-t002:** Detection and absolute quantification of CAstV associated with WCS in the serum, spleen, liver, jejunum, and thymus.

Detection and Quantification of CAstV Associated with White Chicks Condition																					
Organs	Bird 1		Bird 2		Bird 3		Bird 4		Bird 5		Bird 6		Bird 7		Bird 8		Bird 9		Bird 10		Total
	CAstV	VP/mG	CAstV	VP/mG	CAstV	VP/mG	CAstV	VP/mG	CAstV	VP/mG	CAstV	VP/mG	CAstV	VP/mG	CAstV	VP/mG	CAstV	VP/mG	CAstV	VP/mG	VP/mG
Serum	+	1368	+	1884	+	75	+	314	+	409	+	1469	+	99	+	1262	+	662	+	133	7677
Spleen	+	28,364	+	1,173,399	+	1379	+	771,929	+	1174	+	144,512	+	1028	+	530,225	+	192,106	+	104,599	3,928,715
Liver	+	1156	+	354,865	+	2094	+	372,304	+	802	+	84,940	+	1050	+	156,095	+	204,238	+	580,590	1,758,134
Jejunum	+	43,179	+	7,247,165	+	122,799	+	35,037,330	+	127,962	+	48,424,590	+	43,161	+	10,474,623	+	41,159,980	+	58,520,448	201,201,237 **
Thymus	+	4209	+	10,100	+	5292	+	34,351	+	9826	+	6457	+	6642	+	36,293	+	16,006	+	39,373	168,549

Chicken astrovirus (CAstV); viral particles (VP); milligrams (mG); ** = statistically significative difference (0.05) among the tissues from birds affected with WCS.

**Table 3 animals-10-01195-t003:** Comparison of the nucleotide and amino acid identities of the sequences of Brazilian isolates of CAstV with other sequences of this virus.

N.	Groups	Sequences	Percent Amino Acid Similarity																																
			*Ai*				*Aii*				*Aiii*				*Bi*					*Bii*					*Biii*					*Biv*					ANV
			1	2	3	4	5	6	7	8	9	10	11	12	13	14	15	16	17	18	19	20	21	22	23	24	25	26	27	28	29	30	31	32	33
**1**	***Ai***	JN582319.1	-	89.7	99	95.1	77.6	77.4	77.7	78	79.6	79.3	79.3	79.6	38.4	38.4	38.4	38.6	38.6	37.9	37.9	37.7	37.7	37.9	39.2	39.2	39.1	39.2	38.4	38.6	38.6	38.6	38.6	38.6	24.3
**2**		JN582317.1	78.5	-	90.1	90.5	77.5	77.4	77.8	78.1	79.1	78.5	78.5	79.1	38	38	38.2	38	38.1	37.9	38.3	38	38.2	38.2	39.3	39.2	39.2	39.3	38.5	38.8	38.8	38.8	38.8	38.9	24.4
**3**		JN582320.1	97.3	78.7	-	95.5	77.2	77	77.3	77.6	79.5	79.1	79.1	79.5	38.3	38.3	38.3	38.4	38.4	37.7	37.7	37.6	37.6	37.6	39.1	39.1	39	39.1	38.3	38.4	38.4	38.4	38.4	38.4	23.9
**4**		JN582318.1	86.3	77.8	86.7	-	76.3	76.2	76.5	76.7	79.6	79.3	79.3	79.6	38.2	38.2	38.3	38.3	38.3	37.7	37.9	37.7	37.7	38	39.2	39.2	39.1	39.2	38.4	38.7	38.7	38.7	38.7	38.7	24.2
**5**	***Aii***	JN582326.1	70.9	70.7	71.2	70.9	-	99.3	99.1	99.4	81.9	81.7	81.7	81.9	38.6	38.6	38.4	38.7	38.7	37.9	37.1	36.9	36.9	37.1	38.7	38.6	38.6	38.7	37.9	38.2	38.2	38.2	38.2	38.2	23.9
**6**		JN582325.1	70.9	70.6	71.2	70.8	99.7	-	99	99.3	81.8	81.5	81.5	81.8	38.4	38.4	38.3	38.6	38.6	37.7	36.9	36.8	36.8	36.9	38.6	38.4	38.4	38.6	37.8	38	38	38	38	38	24
**7**		JN582324.1	70.9	70.9	71.1	70.8	98.9	98.8	-	99.4	81.9	81.7	81.7	81.9	38.6	38.6	38.4	38.7	38.7	37.9	37.1	36.9	36.9	37.1	38.7	38.6	38.6	38.7	37.9	38.3	38.3	38.3	38.3	38.3	24.2
**8**		JN582323.1	70.9	71.1	71.2	70.8	98.1	98.1	98.2	-	82.2	81.9	81.9	82.2	38.7	38.7	38.6	38.8	38.8	38	37.2	37.1	37.1	37.2	38.6	38.4	38.4	38.6	37.8	38.3	38.3	38.3	38.3	38.3	24.2
**9**	***Aiii***	KT886453.1	71.2	71.1	71.2	71.5	74.8	74.8	75.1	75.4	-	98.3	98.3	100	38	38	38.2	38.2	38.2	37.2	37.1	37.1	37.1	37.5	38.6	38.4	38.3	38.4	38.2	37.8	37.8	37.8	37.8	37.8	24.7
**10**		JN582321.1	71.2	71.1	71.2	71.4	75	75.1	75.3	75.8	97	-	100	98.3	38.2	38.2	38.3	38.3	38.3	37.2	37.1	37.1	37.1	37.5	38.7	38.6	38.4	38.6	37.9	38	38	38	38	38	24.7
**11**		JN582322.1	71.2	71.1	71.2	71.4	75	75.1	75.3	75.8	97	100	-	98.3	38.2	38.2	38.3	38.3	38.3	37.2	37.1	37.1	37.1	37.5	38.7	38.6	38.4	38.6	37.9	38	38	38	38	38	24.7
**12**		KR052479.1	71.3	71.1	71.3	71.5	74.8	74.8	75.1	75.4	99.7	97.2	97.2	-	38	38	38.2	38.2	38.2	37.2	37.1	37.1	37.1	37.5	38.6	38.4	38.3	38.4	38.2	37.8	37.8	37.8	37.8	37.8	24.7
**13**	***Bi***	HQ330506.1	47	47.2	47.1	46.9	46.9	47	47	47.1	46.3	46.5	46.5	46.2	-	100	97.6	98.2	98.2	83	83.1	83.7	83.4	82.6	91.8	91.7	91.3	91.5	91	88.3	88.2	88.3	88.3	88	23.4
**14**		JN582327.1	47	47.2	47.1	46.9	46.9	47	47	47.1	46.3	46.5	46.5	46.2	100	-	97.6	98.2	98.2	83	83.1	83.7	83.4	82.6	91.8	91.7	91.3	91.5	91	88.3	88.2	88.3	88.3	88	23.4
**15**		JN582305.1	47	47.2	47.2	46.8	47.2	47.3	47.3	47.4	46.4	46.9	46.9	46.5	98.5	98.5	-	98.5	98.6	83.4	83.7	84.4	84.1	83.3	92.6	92.5	92.1	92.4	91.8	88.8	88.7	88.8	88.8	88.6	23.4
**16**		JN582311.1	46.8	47.5	47.1	47.2	47.2	47.2	47.3	47.4	46.5	46.9	46.9	46.5	97.3	97.3	97.6	-	99	83.6	83.7	84.4	84.1	83.3	92.5	92.4	92	92.2	91.7	88.8	88.7	88.8	88.8	88.6	23.5
**17**		JN582310.1	46.9	47.5	47.1	47.2	47.2	47.2	47.4	47.4	46.6	46.8	46.8	46.6	97.4	97.4	97.7	99.4	-	83.8	84	84.6	84.4	83.6	92.6	92.5	92.1	92.4	91.8	89.1	89	89.1	89.1	88.8	23.4
**18**	***Bii***	JF414802.1	46.4	46.5	46.5	46.6	46.4	46.4	46.5	46.4	46.2	45.9	45.9	46.1	74.5	74.5	74.2	74.7	74.6	-	94.3	95.2	95	94.7	84.1	84	83.8	84	83.4	84.9	84.8	84.9	84.8	84.8	23.5
**19**		JN582314.1	46.6	46.5	46.5	46.1	45.8	45.9	45.7	46	45.6	45.7	45.7	45.6	74.3	74.3	74.5	74.7	74.7	87.4	-	98.2	98.1	96.6	84.9	84.8	84.6	84.9	84.2	84.8	84.6	84.8	84.6	84.8	23.4
**20**		JN582313.1	47	46.9	47	46.4	46	46.1	45.9	46.2	45.9	46	46	45.9	74.4	74.4	74.7	74.9	74.9	87.6	98.2	-	98.7	97.8	85.3	85.2	84.9	85.2	84.6	85.3	85.2	85.3	85.2	85.2	23.3
**21**		JN582316.1	46.9	46.7	46.9	46.5	46.3	46.3	46.2	46.5	46.1	46	46	46.1	74.3	74.3	74.6	74.9	74.8	87.2	97.6	98.6	-	97.1	85.2	85	84.8	85	84.5	85	84.9	85	84.9	85	23.4
**22**		JN582315.1	46.6	46.7	46.9	46.4	45.8	45.9	45.8	46	45.7	45.7	45.7	45.7	73.7	73.7	73.9	74.1	74.1	86.9	95.3	96.3	96.1	-	84.4	84.2	84	84.2	83.7	84.4	84.2	84.4	84.2	84.2	23.3
**23**	***Bii***	JX945862.1	48.2	47.9	48.2	47.4	47.8	47.7	47.6	47.8	46.6	47	47	46.6	85.3	85.3	85.6	84.9	85.1	75.5	75.7	76	76	75	-	99.8	99.4	99.7	99	89.9	89.8	89.9	89.9	89.8	24.1
**24**		KJ621032.1	48.1	47.8	48.1	47.4	47.8	47.8	47.7	47.8	46.7	47	47	46.6	85.4	85.4	85.5	85	85.1	75.6	75.7	76.1	76.1	75	99.4	-	99.3	99.5	98.9	89.8	89.7	89.8	89.8	89.7	24.1
**25**		KJ621027.1	48.2	47.7	48.1	47.4	47.6	47.6	47.5	47.6	46.7	46.9	46.9	46.7	85.4	85.4	85.5	85	85.2	75.5	75.8	76.1	76.1	75.1	98.3	98.6	-	99.7	98.5	89.5	89.4	89.5	89.5	89.4	24.2
**26**		JX945859.1	48.3	47.7	48.2	47.5	47.6	47.6	47.5	47.6	46.7	46.9	46.9	46.7	85.5	85.5	85.7	85.2	85.3	75.6	75.9	76.1	76.1	75.2	98.3	98.6	99.7	-	98.7	89.7	89.5	89.7	89.7	89.5	24.2
**27**		JX945863.1	47.8	47.8	47.8	47	47.5	47.4	47.3	47.5	46.5	46.8	46.8	46.4	84.9	84.9	85.1	84.5	84.7	75.4	75.5	75.8	75.8	74.8	98.9	99	97.9	97.9	-	89.1	89	89.1	89.1	89	23.7
**28**	***Biv***	USP 748-16	47.5	47.8	47.3	47.4	47	47	47.1	47.1	47	47.1	47.1	46.8	78.5	78.5	78.3	78.8	78.9	77	76.9	77.1	77.1	76	80.6	80.6	80.5	80.5	80.4	-	99.8	100	99.8	99.5	24.1
**29**		USP 748-18	47.5	47.8	47.3	47.3	46.9	47	47	47	46.9	47	47	46.7	78.7	78.7	78.5	79	79.1	77	76.7	77	77	75.9	80.5	80.6	80.4	80.4	80.4	99.7	-	99.8	99.7	99.4	24.2
**30**		USP 748-12	47.5	47.8	47.3	47.4	46.9	47	47	47	47	47.1	47.1	46.8	78.6	78.6	78.4	78.9	79	76.9	76.8	77	77	75.8	80.6	80.7	80.6	80.6	80.5	99.8	99.7	-	99.8	99.5	24.1
**31**		USP 748-14	47.5	47.8	47.3	47.3	46.9	46.9	47	47	47	47.1	47.1	46.8	78.6	78.6	78.4	78.9	79	77	76.6	76.9	76.9	75.7	80.6	80.6	80.5	80.5	80.4	99.7	99.7	99.7	-	99.4	24.1
**32**		USP 748-20	47.4	47.9	47.3	47.4	46.9	47	47	47	47	47.1	47.1	46.8	78.3	78.3	78.2	78.7	78.8	76.8	76.7	77	76.9	75.8	80.5	80.6	80.4	80.4	80.3	99.5	99.5	99.6	99.5	-	23.9
**33**	**ANV**	NC_003790.1	37.2	37.4	37.1	37.3	36.1	36.1	36.4	36.3	37.5	37.5	37.5	37.5	38.7	38.7	38.8	38.9	39	38.2	37.7	37.8	37.9	37.7	39.2	39.2	39.2	39.1	39	39.4	39.3	39.3	39.3	39.2	-
			**Percent Amino Acid Nucleotides**																																

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
