# Peer review of "Molecular Characterization and Determination of Relative Cytokine Expression in Naturally Infected Day-Old Chicks with Chicken Astrovirus Associated to White Chick Syndrome"

_animals, 2020, doi:10.3390/ani10071195_

Round 1
Reviewer 1 Report
Manuscript by Luis Fabian N. Nuñez et al describes molecular characterization and determination of 2 cytokine relative expression in naturally infected 3 day-old white chicks with chicken astrovirus.
The study's objectives are good but need vast improvement in its contents including writing.
The following are my major comments/ concerns:
The manuscript is half built on characterizing the immune responses to astrovirus. The authors mention immune 'modulation' throughout the manuscript. The study has tested many cytokines gene expression that represent pro- and anti-inflammatory phenotypes and TH1/Th2 etc. However,
- it seems like they have pooled the tissue samples. Was this due to scarcity of adequate samples from day-old chicks? If yes, it could still be pooled in a way to make representative samples (like pool 2-3 per sample) and tissues like jejunum would have been dealt individually. The figure therefore has variable features to represent and discuss.
- Were there healthy controls used in immune gene expression? If yes, why does the figure shows only infected tissue expression. Was the control samples expression used as a background to substract the expression in infected tissues? If yes, how exactly was it done? Cite relevant references.
- There needs to be a systematic evaluation of cytokine expression to craft out the discussion, which is very poorly written.
- Based on cytokine expression data, the conclusions seem over estimated and sometimes appear speculative.
I have made numerous comments in the manuscript to help improve its strength and pl see it attached.

Author Response
Dear Reviewer, please see attached the comments that were added in text, according your comments and suggestions. I thank you very much for your efforts in becoming our manuscript much better and suitable for publication in Animals. Best regards.

Reviewer 2 Report
Hi dear authors,
This is very interesting subject highlighting the connection of Astroviruses (AstV) to a chick syndrome; so called White Chick Syndrome (WCS), resulted in boosting the importance of AstVs more than before in veterinary science. The paper is novel to describing the evidence for this connection. Some comments, however, it is thought to be suggested to improve this paper for a better understanding. Here is the list of comments:
1. Introduction
Major comment
There would be good to add some sentences about the morbidity and mortality rate of WCS in poultry industry, when the chicks get the CAstV infection, the route of transmission of CAstV and other epidemiological aspects (risk factors) of WCS and CAstV.
Minor comments
Line 63: "hatching [8,18]" instead of "hatching 8,18]" would be nice.
2. M&M
Major comment
It would be nicety add the ethical permission number and animal welfare association for the using these chicks for this research purpose (Line 98). So the research would be ethically approved and acceptable by the public and law enforcement association.
Minor comments
Line 164: "was generated here and presented as viral gene copies per mg" would be nice to be added.
Line 182: "other sequences deposited in GeneBank": it would be nice to add a "supplementary Table" for all these "other sequences" describing their Identity, year of isolation, origin, region of isolation and any other useful information that can help the readers to understand better the "Table 3" and the phylogenetic tree in "Figure 5".
3. Results
Major comment
Figures illustration can be improved to better helping the text. More details should also be added to the legends of the figures for better connection of the legends to figures.
Minor comments
Line 215: bring another research example to support that "ten chicks" was enough to prove the WCS-CAstV connection for this study. This "ten" was from how many chicks in total? And is this ratio normal for WCS, reported else where?
Line 219: since later you used "Yolk sac", it would be nice to change "vitelline sac" to "yolk sac".
Lines 226-228 (Figure 1): seems B and C are from same chick but different magnification. If so, please keep one of them (I suggest figure 1C). Indeed, it would be nice to use arrows to show the tissues and organs in the Figure 1C to connect the figure better to the legend of the figure.
Lines 229-231 (Figure 2): It would be nice to add "A" and "B" section in to the legend of the figure with a short description for each section. Indeed, it would be nice to use arrows to show the tissues and organs in the Figure 2 to connect the figure better to the legend of the figure.
Lines 240 (Figure 3): It would be nice to label the figure sections with "A" and "B", instead of "Left" and "Right".
Lines 241 (Figure 3): it would be nice to add which part of the genome of CAstV was amplified.
Lines 242 (Figure 3): "had an efficiency" or "had a sensitivity"or "had a specificity"? Please clarify "efficiancy".
Lines 247 (Table 2): "absolutive" or "absolute"?
Table 2: it would be nice to add a footer description about sign"*".
Table 2: in Jejunum section, for chicks 4,6,8,9,10 and total, there are two numbers. What is the second number? There is big difference between two numbers too. It would be nice to clarify this two-numbers section.
Lines 250-254 (Figure 4): It would be nice to add some description about the "method of analysis" to obtain these data, which I assume it should be RT-QPCR.
Lines 252 (Figure 4): "Monk" or "Mock"?
Figure 4: there is header title on the top of Figure. It would be better to transfer it to the legend of the figure in line 250.
Figure 4: the selected figure legends to illustrate each column is a bit confusing. It would be nice to chose other legends for each column to distinguish the columns easier from each others.
Lines 273 (Figure 5): "infected with WCS" or "affected/complicated with WCS"? Indeed, it would be nice to also "highlight the other Brazilian CAstV clustered in group Bii" with another color in the phylogenetic tree. Then please add their descriptions in thelelgend of the figure to better understand the tree.
Lines 292-295 (Figure 6): It would be better to summerize all these four sections in "one Heatmap" Figure. The "Figure 4" in paper "Bidokhti, et al. 2019. Immunogenicity and Efficacy Evaluation of Subunit Astrovirus Vaccines. Vaccines 2019, 7, 79." from similar research can be used as an example.
4. Discussion
Major comment
The text should be improved and more updated references should be used. A lots of sentences are connected via ";" which is confusing. It would be nice to replace it with ".", for example Line 347 "[15]; in the present work" would be better to change to "[15]. In the present work". Indeed, making paragraphs would be nice to be improved. Some consecutive paragraphs look so connected and should become one paragraph.
Minore comments
Line 297: adding "(CAstV)" after "Chicken astrovirus" would be nice. Indeed, is this "chick" or "chicken"?
Lines 301-307: it would be nice to add some 1-2 sentences. about CAstV morbidity and mortality rates and way of transmission, with references.
Line 308: "by our working group" in which year on how many chicks? morbidity and mortality rates. It would be nice to add more description of your previous work research here.
Line 318: "with other previous reports", please add the references here, instead of end of paragraph [8,17].
Line 317: "Liver" or "intestine"? the text would be nice to be improved.
Line 319: "[21]" is this your finding or from reference 21? it would be nice to clarify the text.
Line 325: "chicken" or "chick"?
Line 331: it would be nice to add some examples of "previous molecular techniques" such as.
Line 332: "WCS" would be better to be removed.
Line 333: "in several organs" it would be nice to add reference for this sentence. Indeed, using "." instead of ";" would be better.
Line 338: what is the source of CAstV infection. Does this line means that chicks get CAstV from either "their moms" or "in hatching machine after birth"? please add some words about where is this infection originated? plz add references, if any research has been done.
Line 342: The source of this high viral load in liver could be "remaining blood in the liver". Have you done PBS perfusion through the heart puncture to remove the blood from tissues before tissue sampling? If so, please add some sentences about this PBS perfusion to the M&M, section "2.1.".
Line 345: "WCS sequences" or "CAstV sequences"?
Line 348: "in GeneBank; interestingly," to "in GeneBank. Interestingly," would be better.
Lines 353-355: the text would be nice to be improved with better words.
Line 361-362: "the jejunum" would be nice to change to "jejunum".
Lines 362 and 375: "despite the high virus concentration" would be nice to be replaced by "despite of high viral loads". This has been seen in other parts of the papers too which it is suggesting to edit it too.
Lines 362-366: this has been well described in other research. it would be nice to update the references here by adding "Bidokhti, et al. 2019. Immunogenicity and Efficacy Evaluation of Subunit Astrovirus Vaccines. Vaccines 2019, 7, 79." and "Krishna NK. Identification of structural domains involved in astrovirus capsid biology. Viral Immunol. 2005;18(1):17‐26. doi:10.1089/vim.2005.18.17". In both of these papers, they describes the immunpsuppressing activity of the AstV, leading to viral replication in the presence of high IFN-y and TGF-b levels.
Lines 367-377 (paragraph 1) is so connected to Lines 378-384 (paragraph 2); both describing the immunpsuppressing activity of AstV, leading to high virus replication. Thus, it would be nice to connect these two paragraphs in to one paragraph.
Lines 360-366: did you test the tissue samples for other pathogens as well? Did you test any chick with WCS for negative CAstV? Maybe other pathogens could cause this WCS too.
Line 368: "as previously mentioned" would be better to be removed.
Lines 372-374: "This condition also appears in the present investigation in the jejunum, where there is gene expression of cytokine TGF- β but also significant expression of IL-15," would be nice to replaced with "In jejunum, there is gene expression of TGF- β and IL-15". Such editing sentences is highly recommended to improve this interesting paper.
Lines 379-380: "but also protects against HAstV-1 increased barrier permeability" is unclear. it would be nice to clarify this sentence.
Line 379: extension of abbreviated "Caco-2" would be nice to be added.
Lines 385-388: In which animals was this study done? please mention them here.
Line 395: extension of abbreviated "FAdV-4" would be nice to be added.
Lines 355-396: In which animals was this study done? please mention them here.
Overally, this is an interesting paper and can be considered for publication after editing and revising using the comments kindly dedicated above.
Thank you very much for nice research and Good Luck
Author Response

(The authors gave the same response as above.)

Round 2
Reviewer 1 Report
The revised manuscript reads well and the scientific quality has been significantly improved. I have no further comments/suggestion/changes.